# Brief communication: Preliminary hydro-meteorological analysis of the flash flood of 20 August 2018 on "Raganello Gorge", Southern Italy

Elenio Avolio[1], Ottavio Cavalcanti[2], Luca Furnari[2], Alfonso Senatore[2], Giuseppe Mendicino[2]

[1]Institute of Atmospheric Sciences and Climate - National Research Council (ISAC-CNR), Lamezia Terme, 88046, Italy
[2]Department of Environmental and Chemical Engineering, University of Calabria, Rende, 87036, Italy

*Correspondence to*: Alfonso Senatore (alfonso.senatore@unical.it)

**Abstract.** On 20 August 2018 a flash flood affected the Raganello Creek (Southern Italy) causing 10 casualties. The rainfall
event was so highly localized that the spatial coverage of rain gauges resulted inadequate to measure it, while radar products showed a storm cell with rain peaks of about 70-100 mm/h. This scientific report provides a preliminary hydro-meteorological analysis of the event and evaluates the forecasting skills of a system based on the WRF/WRF-Hydro models, using both one-way and fully-coupled approaches. First results show a reasonable simulation of the event in terms of both rainfall and hydrological impact.

## 1 Introduction

On 20 August 2018, in the early afternoon, a flash flood affected the downstream outlet of the gorge of the Raganello Creek Catchment (Calabria region, Southern Italy), causing the death of 10 hikers and the wounding of as many people. The event, which is currently subject of a judicial inquiry, has had considerable media coverage, surely due to the loss of human lives but also for the debate regarding possible responsibilities in the emergency management.

The main challenge for the scientific community studying extreme atmospheric events is to develop reliable modelling systems able not only to reconstruct the events, to understand their characteristics and dynamics, but also to timely forecast the possible effects, in order to implement feasible mitigation actions for damages to people and infrastructures. From this point of view the study of the Raganello event, though challenging due to its characteristics of very high spatial-temporal localization, is of particular interest. This study represents the first scientific analysis of the flood.

Calabria region is particularly prone to extreme precipitation events. Many studies conducted over the years investigated the hydro-meteorological causes of several flash-floods affecting this fragile territory (Federico et al., 2003a, 2003b; Gascòn et al., 2016). A major flood event for Calabria was analysed in a recent work (Avolio and Federico, 2018), where different configurations of the WRF model (Skamarock et al., 2008) were tested and various sensitivity tests were performed, with the aim of identifying the best model configuration. WRF is a state-of-the-art high-resolution mesoscale atmospheric model
system that was used also in this study (in the version 3.9.1) together with its hydrological extension WRF-Hydro (version 5.0; Gochis et al., 2018), adopting both one-way and fully-coupled approaches.

Several WRF simulations were carried out before choosing the best configuration, to test the impact of the initial conditions, horizontal resolutions and parameterization schemes. Based on the best-choice WRF configuration, the hydrological module was activated in order to provide a streamflow forecast at the outlet of the analysed catchment.

The main objective of this work is to study the event from a hydro-meteorological point of view, and to analyse the causes responsible for its high impact, with the ambition of providing useful preliminary indications concerning the best possible use of the WRF model for hydro-meteorological forecasting purposes in the study area.

## 2 Data and methods

The event was characterized by a high spatial-temporal localization. For such small-scale events the available observations have proved to be insufficient because the regional rain gauge network, maintained by the "*Centro Funzionale Multirischi*" of the Calabrian Regional Agency for the Protection of the Environment (http://www.cfd.calabria.it), was not dense enough in the surroundings of the site of interest. Also, at the time of writing, the Civil Protection radar images are not fully available due to on-going judicial investigations.

A source of potentially useful data is related to a technical report of the event carried out by the "*Centro Funzionale Multirischi*" (Centro Funzionale Multirischi della Calabria, 2018), where a rainfall analysis and an image of the Surface Rainfall Total (SRT) spatial distribution, derived from the national weather radar network managed by the Civil Protection Department, are provided; the SRT estimation is obtained according to nine steps detailed in Vulpiani et al. (2014), Petracca et al. (2018) and references therein.

Despite the scarcity of available observations, we will base our modelling analysis taking into account the few rainfall measured data and the information reported in the aforementioned technical report, in particular the SRT image of estimated 3-hour precipitation (Fig. 2 of the report; Centro Funzionale Multirischi della Calabria, 2018). This integrated approach, that allows to consider both rain gauges and radar products, represents a more complete basis for the analysis.

### 2.1 Description of the event

On 20 August 2018 an intense rainfall hit the northern part of the Calabria region, and specifically the Pollino Mountain, where the Raganello Creek Catchment is located. This peak is one of the highest in southern Italy, exceeding 2000 m AMSL. Its complex and particularly steep orography makes it difficult to easily understand all the involved physical factors and their relative contribution to the event development.

Most of the rain affected the region (Fig. 1), in particular the western (Tyrrhenian) side, starting from the second half of the day. In the study area the accumulated precipitation was highly localized, mainly affecting the north-western part of the Raganello Creek Catchment, while the flood wave that caused the fatalities occurred downstream, more to the south-east, near the town of Civita (Fig. 1b). At that outlet, the catchment extent is about 100 km$^2$ and the streamflow is perennial. According to the Corine Land Cover 2018 inventory, almost half of the land is covered by forest (44%, almost all broad-

leaved), 22.9% by shrubs, 21.8% by agricultural areas (13.8% heterogeneous agricultural areas, 7.7% non-irrigated arable land), 11% by open spaces with little or no vegetation. Artificial surfaces are only 0.3%.

Most probably, the destructive power of the flood wave was amplified by passing for about 12 km through a narrow gorge with an average slope of about 30%. Nevertheless, since there are no available gauge stations along the river, no discharge neither water levels were measured; currently, the consultants appointed by the judicial authority are indirectly reconstructing them.

Figure 1b shows the study area with the available rain gauges and the precipitation recorded during the whole day. Figure 1c shows the 24h accumulated precipitation simulated by WRF, in its better configuration (see in the following). The model simulates a moderate/high amount of rain, mainly to the northwest part of the catchment outlet at Civita (identified by a red (black) dot in Fig. 1b (1c)).

## 2.2 Large scale conditions

For the sake of conciseness, the maps related to the large-scale discussion are not shown in this brief communication; some figures are reported in the Supplement (Section S1).

The synoptic analysis (Fig. S1 in the Supplement) reveals, at 850 hPa, the presence of a trough moving from Sardinia to Sicily (from NW to SE); this low was associated with a core of relatively cold air at medium-high altitude. The related currents had a cyclonic circulation around the Calabria and hot and potentially unstable air masses, coming from E-SE, were advected toward the study area. The satellite images of the thermal infrared channel (10.8 μm) (Fig. S2 in the Supplement) confirmed this configuration, showing the presence of clouds and storm cells in correspondence of the low-pressure area, i.e. in the southern Tyrrhenian Sea, and, successively, of the western coastal zones. In correspondence of these cloudy systems a large number of strokes (Fig. S3 in the supplement), recorded by the LINET network (Betz et al., 2009), further confirmed the deep convective activity of this event.

## 2.3 Modelling strategy: approach and tools

Several WRF simulations were carried out before choosing the best configuration, in order to test the impact of the initial conditions, horizontal resolutions and parameterization schemes. Ten different combinations were preliminarily tested, but they are not discussed in detail in this brief communication; Table S1 in the Supplement summarizes these tests and a brief comment about them is provided.

Given the limited data available, especially in the small area of the Raganello Creek Catchment, no traditional scores were calculated, but a quantitative/qualitative comparative analysis was carried out to identify the WRF configuration able to simulate more realistically the rainstorm. The choice was performed comparing the precipitation field estimated by the radar (SRT) and that simulated by WRF, in terms of better quantitative/qualitative representation of the rain near the site of interest. Some information about the choice of the optimal configuration is provided in the Supplement Material (Fig. S4 and the short comments related to it, Section S2).

This first comparison allowed identifying a basic configuration (*RUN_SA*) simulating reasonably well the event, adopting two nested (one-way) grids, shown in Fig. 1a. The first domain D01 covers the central Mediterranean basin (6 km grid spacing in both NS and WE directions; $312 \times 342$ grid points); the second domain D02 represents the Calabrian Peninsula (2 km grid spacing; $200 \times 200$ grid points). The model was implemented with 44 terrain-following vertical levels. A 24-h run was performed, starting at 00 UTC of 20 August 2018; since the event was recorded in the early afternoon, this choice allows a reasonable spin-up time. The ECMWF's Integrated Forecasting System (IFS), in its deterministic forecast version at 9 km resolution, provided initial and boundary conditions.

The physical parameterizations adopted were: the Rapid Radiative Transfer Model (RRTM) long-wave radiation scheme (Mlawer et al., 1997), the Goddard shortwave radiation scheme (Chou and Suarez, 1994), the Unified NOAH Land Surface Model (Tewari et al., 2004), the New Thompson microphysics scheme (Thompson et al., 2008), the Mellor–Yamada–Janjic (MYJ) scheme (Janjic and Zavisa 1994) for the Planetary Boundary Layer (PBL) and the Tiedtke cumulus parameterization (Tiedtke, 1989) scheme activated only for the coarser grid. Furthermore, the *sst_update* option, allowing dynamical lower boundary conditions, and the *sst_skin* option (Zeng and Beljaars, 2005), permitting to take into account Sea Surface Temperature (SST) dynamics, were used.

Once the basic configuration was chosen, two further approaches were tested.

The first new test (*RUN_FC*) was a fully-coupled atmospheric-hydrological approach by means of the WRF-Hydro extension (Gochis et al., 2018). Essentially, WRF-Hydro provides a coupling architecture allowing to connect the processes simulated by the meteorological model to lateral surface and sub-surface water flows modelled by a higher resolution hydrological model. Such connection can be both one-way, i.e., like a 'classical' meteo-hydrological forecasting chain where the output of the meteorological model is used as input, or two-way, with a feedback from the routing models to the atmosphere. The WRF-Hydro modelling system has been already used in Calabria, both one- and two-way (Senatore et al., 2015). In the *RUN_FC* simulation WRF-Hydro run in two-way mode, in order to check possible further improvements of the forecast. The terrain file needed for the hydrological analysis has a resolution of 200 m (hence, a disaggregation factor of 10 with respect to the innermost WRF domain was applied). Though the hydrological analysis is focused only over the Raganello catchment, such file must have the same extension of the innermost domain, so the grid dimension is $2000 \times 2000$ grid points. Initial conditions for the hydrological model were provided by ECMWF's IFS. Since there are no streamflow data available, the default parameters were retained with only one change concerning the surface runoff parameter (REFKDT), which was halved from the original value of 3.0 to 1.5. This change was made in order to compensate for the more frequent calls to the NOAH vertical infiltration scheme (in the order of seconds) occurring when the fully coupled option is adopted, because the lateral redistribution of water allows more infiltration. Furthermore, it agrees with previous calibrations in neighbouring catchments (Senatore et al., 2015).

The second approach was aimed at an improved representation of the initial and lower boundary skin SST conditions, and was tested with both one-way (*RUN_SST_SA*) and two-way coupling (*RUN_SST_FC*). Several studies have shown that SST representation in coastal areas (and specifically in Southern Italy; Senatore et al., 2014) can affect significantly the resulting

precipitation fields. In this specific case, native IFS SST fields are affected by an interpolation problem along coastlines that lowers temperatures to unrealistic values, close to 0 °C (L. Magnusson, personal communication). Such problem was addressed in the pre-processing phase of the simulation in a simple but effective way by means of few GIS-based operations, replacing clearly unrealistic values, which were filtered by means of a percentile approach, according to a nearest neighbour rule.

## 3 Results and discussion

The flood wave causing the disaster occurred at about 13 UTC. Fig. 2a shows the cumulated 3h-rain estimated by the weather radar (the only available observation of the main rain event) in the period from 10 to 13 UTC (Centro Funzionale Multirischi della Calabria, 2018). This image is directly comparable with Figs. 2b-e, which represent the 3h precipitation simulated by WRF, for the same time interval, with the four different approaches adopted. The radar estimate in Fig. 2a identifies large areas affected by medium/high precipitation values, with a highly localized maximum over the northern boundary of the catchment. The precipitation pattern, according to this image, had a particular "C" shape, and involved other mountainous areas of the Pollino range and north-western coastal areas of Calabria.

Figure 2b (*RUN_SA*) refers to the basic configuration chosen after the aforementioned preliminary comparisons. The 3h precipitation simulated by the model appears in good qualitative agreement with the radar estimate. The "C" shape of the rain pattern is reasonably well simulated, resulting only a little more elongated than the radar estimate; the rainfall in the northern Tyrrhenian coastal zone is correctly reproduced and high rainfall zones (> 60 mm/3h) are simulated thereabout of the study area. The main maximum is located at about 17 km northwest with respect to radar estimates. The rainfall value is lower, because the highest precipitation is forecasted by WRF with about one-hour delay (rainfall peaks equal to 59 mm/3h and 83 mm/3h, in the time intervals 10-13 UTC and 11-14 UTC respectively; similar delay is experienced for all simulation tests). Overall, this basic model configuration shows reasonable results, although the precipitation field suffers from some displacement/underestimation errors. Furthermore, simulated rainfall underestimation with respect to the SRT image is exaggerated by the small forecast delay.

Figures 2c (*RUN_FC*), 2d (*RUN_SST_SA*) and 2e (*RUN_SST_FC*) refer to the three further simulation tests performed.

The differences between *RUN_FC* and *RUN_SA* are small, because the main effects of the fully coupled approach are expected to be mainly given by a different soil moisture distribution, but the short-time simulation did not allow it to evolve significantly. A possible modelling improvement, which will be tested in future work, is to run the hydrological model off-line for a couple of months ahead, in order to get more detailed initial soil moisture conditions than those provided by the general circulation models.

The improvement provided by *RUN_SST_SA* (Fig. 2d) is instead more significant. The peak of rainfall is higher (plausibly due to the energy surplus passed to the boundary layer by the warmer "corrected" SST near the coastline), and occurs no more to the north of the catchment, but where a secondary maximum was previously located, closer to the north-western

catchment boundaries (about 15 km from the radar estimate maximum). Even though this behaviour does not reproduce perfectly the radar estimates, it provides a more realistic representation of the rainfall over the Raganello catchment.

Finally, such as for *RUN_FC* and *RUN_SA*, also the differences between *RUN_SST_FC* and *RUN_SST_SA* are very small. Nevertheless, from a theoretical point of view the configuration *RUN_SST_FC* simulates more accurately the local water cycle, which indeed is the result of 'fully coupled' processes. This assumption is supported, in this specific case, by a quantitative statistics performed taking into account the 24h-accumulated precipitation measured by the available rain gauges (Fig. 1b). Considering the Mean Bias (MB) and the Root Mean Square Error (RMSE) (Wilks, 2011) for the four configurations described above, the *RUN_SST_FC* is that with lower errors. Specifically, the values of the RMSE (MB) are: 8.0 mm/24h (3.9 mm/24h) for *RUN_SSF_FC*; 10.2 mm/24h (6.1 mm/24h) for *RUN_SST_SA*; 11.7 mm/24h (7.5 mm/24h) for *RUN_FC*; 11.8 mm/24h (8.0 mm/24h) for *RUN_SA*. Configuration *RUN_SST_FC* also provides slightly higher values of rainfall in the catchment during the 24-hour simulation period. In particular, considering the 24h simulated catchment-averaged rainfall (the 24h rainfall peak), we obtain the following results: 20.7 mm (52.3 mm) for *RUN_SA*, 23.3 mm (53.0 mm) for *RUN_FC*, 33.4 mm (81.0 mm) for *RUN_SST_SA* and 40.4 mm (83.1 mm) for *RUN_SST_FC*.

The overall analysis suggests to choose *RUN_SST_FC* as the reference simulation for the following meteo-hydrological analysis of the event.

## 3.1 Mesoscale analysis and hydrological results

Figure 3 shows some maps derived by the WRF *RUN_SST_FC* simulation, describing the behaviour of some significant parameters during event occurrence.

Figure 3a shows the simulated reflectivity at 11 UTC, at the beginning of the three-hour interval represented in Figure 2. Although no reflectivity radar maps were released, comparing the simulated reflectivity with the SRT estimate directly derived from radar (fig 2a) shows that the model reproduced well the perturbation, allowing an ideal overlap with the rain pattern estimated by radar, both in terms of shape and areas involved.

From a meteorological point of view, the most probable cause of triggering for this event is the development of high atmospheric instability in the study area, most likely due to the contrast between hot and humid air masses coming from the East, and the relative cold air associated with the upper-level trough moving from NW (Fig. S1 in the Supplement). The development of vertical instability was further facilitated by the orographic lift of the moist maritime air masses. Also, the surface warming in the morning further favoured the instability of the PBL.

Figure 3b shows the vertical wind speed at 700 hPa at the time of event occurrence (13 UTC). Large areas with high vertical velocities are evident, both positive (updraft) and negative (downdraft), which highlight the development of intense convective motions in the surrounding areas.

A further confirmation of the instability in the area is given by Figure 3c, where the vertical wind shear between 0 and 3 km at the same time of the previous map is shown. In the surroundings of the disaster site values up to 22 m s$^{-1}$ were simulated, indicative of a situation where the development of long-lived multi-cell or supercell storm is favoured.

The k-index (Fig. 3d) is useful for assessing the potential for atmospheric instability in the area; values locally higher than 40 °C were simulated at 13 UTC, suggesting a "high convective potential" situation (George, 1960). The simulated 10 m wind field is superimposed on the k-index map, highlighting several zones of convergence/divergence of surface currents, mainly related to the complex orography in the area.

Finally (not shown for brevity), a high maximum CAPE (convective available potential energy) was simulated in the area, with values locally exceeding 2700 J kg$^{-1}$. Values higher than 2000 J/kg are often associated with strong instability conditions and supercells/tornadoes development (e.g., Rasmussen et al., 1998).

As for concerns the hydrological impact of the event, figure 3e shows both the hourly averaged rainfall values over the catchment and the resulting hydrograph for the *RUN_SST_FC* simulation. It is recalled that, since no information (nor direct neither indirect) about observed streamflow is currently available, no calibration was performed and the default parameters were retained, with the only exception of reduced REFKDT. The same figure shows, as a reference, the hydrograph reconstructed exploiting the available SRT image. The radar-derived precipitation was transformed in WRF-Hydro input assigning to each pixel of the radar image a rainfall value equal to the central value of the corresponding colour class, then the pixels were aggregated from radar (1 km) to WRF (2 km) resolution by calculating the average. Temporal disaggregation was assumed proportional to the WRF output and, since WRF-Hydro was executed off-line, REFKDT was set equal to 3 (therefore, in this simulation the default parameters were completely retained).

Both hydrographs highlight the extremely impulsive response of the catchment to precipitation, with a very small lag time; in order to allow a more accurate description of the event, future work will focus on sub-hourly time scales. The highest hourly rainfall intensity averaged over the catchment was forecasted from 13 to 14 UTC with about 32 mm h$^{-1}$ (with the reconstructed peak of 36 mm h$^{-1}$ occurring one hour before) and a resulting peak flow of about 123 m$^3$ s$^{-1}$ (with the acknowledged one-hour delay with respect to the reconstructed 113 m$^3$ s$^{-1}$). Interestingly, with similar input rainfall amounts, comparable hydrographs are produced by the one-way radar driven and the fully-coupled simulations, provided that the REFKDT parameter is considerably reduced in the latter. This outcome corroborates the strategy adopted for WRF-Hydro parameterization, based on the findings of Senatore et al. (2015).

Due to the lack of a calibration process, both the reconstructed and forecasted peak flows are to be considered only as approximate. Nevertheless, the forecast is still useful if able to give indications for early warning purposes. The forecast and warning lead time of the system mainly depends on the waiting time for the GCM forecast, which in the case of IFS is about six hours. Then, the time needed to run the regional model depends on the characteristics of the local cluster. We performed one-day simulation in less than one hour. Therefore, in this case at about 07 UTC hydro-meteorological forecasts of 00 UTC would be completely available, providing a warning lead time of at least 6 hours. Nevertheless, it is noteworthy to highlight that this evaluation is based on a single-case test. To provide more general indications about its reliability, the forecasting system should be tested for a broader range of events in the region.

As a further evaluation of the reliability of the impact forecast, the peak flow was used to perform a preliminary one-dimensional steady flow simulation at the catchment outlet. Data from an ultra-high resolution (5 m) Digital Terrain Model

provided by the Calabria Region Cartographic Centre made up for the lack of accurate cross-sections measurements, allowing to draw about 70 cross-sections upstream and 10 downstream the catchment outlet, approximately spaced 30 m. The simple hydraulic analysis provided flow velocities ranging from 2.2 to 5.3 m s$^{-1}$ and water levels from 2.0 to 3.3 m in the surroundings of the Raganello gorge outlet. These results roughly agree with water levels of about 2.5 m reported by some

witnesses cited by press (e.g., https://www.corriere.it/cronache/18_agosto_21/gole-raganello-ecco-cosa-successo-8c86c570-a504-11e8-8d66-22179c67a670.shtml, in Italian, last access 1 March 2019), confirming the capability of the system to provide timely information (the simulation time of the hydrodynamic model is a few seconds) on the upcoming flood scenario and activate warnings.

## 4 Summary and outlook

This brief communication is the first scientific report regarding the Raganello flash flood of 20 August 2018, performed throughout the only use of operational forecast models. According to the limited amount of observations currently available, the coupled meteorological/hydrological modelling approach carried out with the WRF/WRF-Hydro modelling system showed its ability to simulate reasonably well the rain field and other important parameters for understanding the meteorological characteristics of the event and forecasting the related flood scenario. The modelling approach permitted to

highlight the major meteorological factors responsible for the development of high atmospheric instability in the area. An important role was played by the contrast between humid maritime air masses and cold upper level ones, as well as by the orographic forcing, in terms of induced lifting and convergence/divergence lines in the area. The improvement in the SST field representation was particularly significant, contributing to a better description of the convective forcing of maritime origin, which affected positively the simulation. The change from stand-alone to fully-coupled modelling was less relevant,

mainly due to the reduced time extension of the simulation.

This first assessment study provides clear indications about the potential predictive capability of a state-of-the-art atmospheric-hydrological modelling system even for very localized events. For a deeper understanding of the physical causes of the event and to further improve the actual skills of the forecasting system, it is necessary to carry out further work, partly already in progress. In particular, the activities underway, or immediately to be carried out, will concern the realization

of further WRF sensitivity tests to provide in-depth physical information supporting the obtained results.

Once full radar data will be utilizable, specific methodologies to improve the forecasts will be performed, particularly by means of variational assimilation techniques (3D-var). Also, an ensemble approach can be useful to improve the prediction of such catastrophic events.

Concerning the hydrological impact, future work will be targeted at analyses at a sub-hourly time step; also, the use of

reconstructed streamflow data (if/when available) allowing the hydrological model calibration will certainly provide further improvements. Furthermore, a specific hydrological analysis assuring a proper spin-up time will be dedicated to initial soil moisture conditions. Accurate soil moisture initial conditions can be achieved operationally through the off-line seamless run

of the hydrological model fed by real-time observations, but this procedure requires that the forecasting system is expressly set-up with several non-trivial procedures. Preliminary land surface spin-up experiments (not shown) obtained executing a simulation with ERA5 reanalysis boundary conditions for the two weeks preceding the event, seem to suggest that improved soil moisture initialization alone is not able to improve convective cell positioning, since the accumulated rainfall fields over the catchment are quite similar, such as the resulting hydrographs. On the other hand, some sensitivity analyses performed perturbing initial soil moisture conditions of simulations RUN_SA and RUN_SST_FC (with uniform changes of ±5%) highlight potentially deep effects on the rainfall peak values. E.g., for RUN_SA, 5% increased initial soil moisture provides enough water to the system, so that the rainfall peak value increases up to a value comparable to RUN_SST_FC. However, with the configuration RUN_SST_FC the rainfall increase/decrease is not so straightforwardly correlated to soil moisture increase/decrease. These results, within the framework of a comprehensive ensemble forecast, will be considered for further improvements of the forecasting system.

**Data availability**. Rainfall data are provided, upon request, by the "Centro Funzionale Multirischi – ARPACAL" (http://www.cfd.calabria.it/) and the Centro Funzionale Decentrato Basilicata (http://centrofunzionalebasilicata.it). The technical report about the Raganello event is freely available at the web address http://www.cfd.calabria.it//DatiVari/Pubblicazioni/rapporto%20evento%2020%20agosto%202018.pdf; (last access: 1 March 2019).

**Author contribution.** EA, AS and GM conceptualized the study. EA and AS developed the methodology and wrote the original draft of the paper. EA, OC and LF carried out the simulations and performed the statistical analyses with the support of AS. LF prepared the figures with the support of EA and AS. GM supervised the research activity and reviewed the original draft.

**Competing interests.** The authors declare that they have no conflict of interest.

**Acknowledgements**. We thank the "Centro Funzionale Multirischi" of the Calabrian Regional Agency for the Protection of the Environment and the Centro Funzionale Decentrato Basilicata for providing the observed precipitation data. LINET data were provided by Nowcast GmbH (https://www.nowcast.de/) within a scientific agreement between H.-D. Betz and the Satellite Meteorological Group of CNR-ISAC in Rome. L. Furnari acknowledges support from the Programme "POR Calabria FSE/FESR 2014/2020 – Mobilità internazionale di Dottorandi e Assegni di ricerca/Ricercatori di Tipo A" Actions 10.5.6 and 10.5.12. Special thanks to Dr. Stefano Federico for the useful discussions and advises.

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

**Figures**

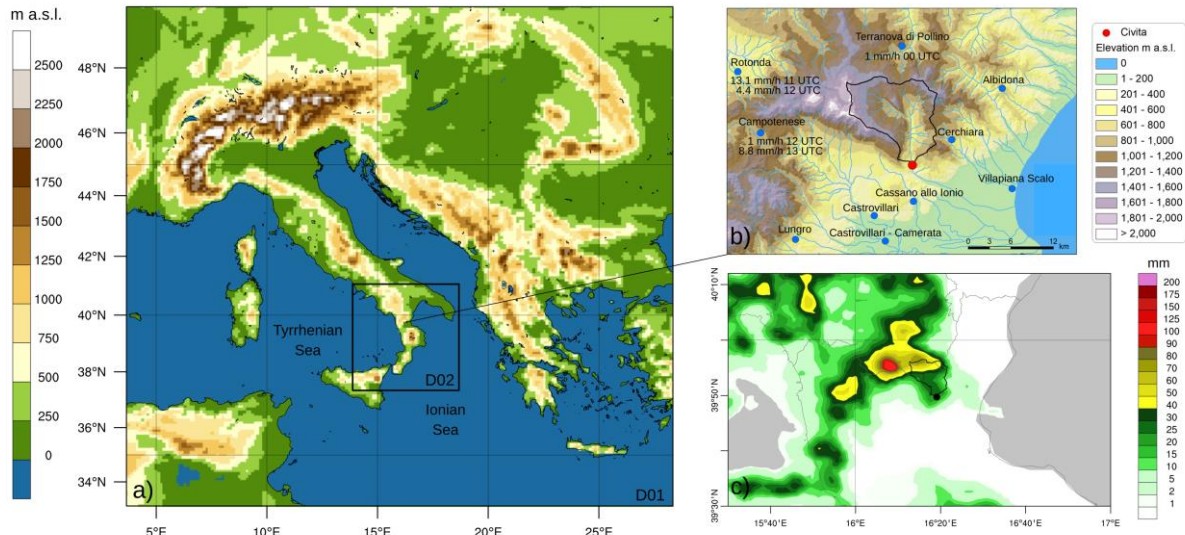

Fig. 1: a) topographic map of the Italian Peninsula in the Central Mediterranean; the map also represents the two grids adopted for the numerical simulations; b) a zoom on the northern part of the Calabria region and on the Raganello Creek Catchment (boundaries in black); the map shows the orography and the available rain gauges with the related 24h precipitation recorded; only for the rain gauges where rainfall was recorded, the related values are indicated. A red dot identifies the site of the disaster, near the town of Civita; c) the 24h cumulated precipitation (mm) simulated by WRF (a zoom on the second grid of the model), according to the run RUN_SST_FC (refer to text for details). The boundaries of the Raganello Creek Catchment are highlighted (black line); administrative borders are also shown in light grey, while the disaster site is identified by a black dot.

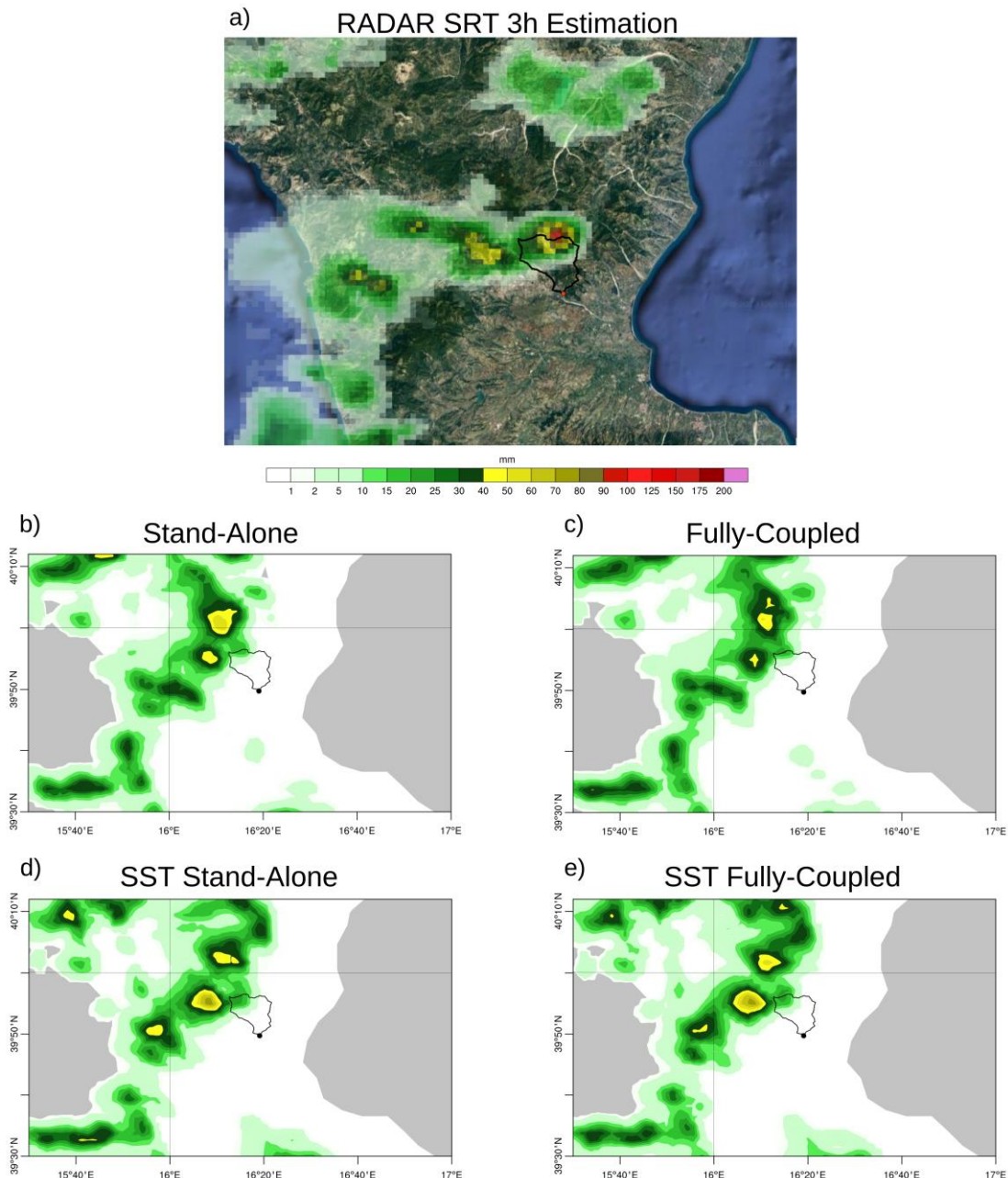

**Fig. 2: a) 3h (10-13 UTC) Surface Rainfall Total (SRT) product derived by radar, provided by 'The Regional Agency for the Protection of the Environment of Calabria region' (source: http://www.cfd.calabria.it//DatiVari/Pubblicazioni/rapporto%20evento%2020%20agosto%202018.pdf, last access 1 March 2019); the map was slightly modified adding the contours of the catchment and the point of the disaster site (red dot); b) 3h (10-13 UTC) accumulated precipitation simulated by WRF for the run RUN_SA; c) as in b) for the run RUN_FC; d) as in b) for the run RUN_SST_SA; e) as in b) for the run RUN_SST_FC. The same colour-bar is used for all the maps. In Figs. from b) to e) the disaster site is identified by a black dot.**

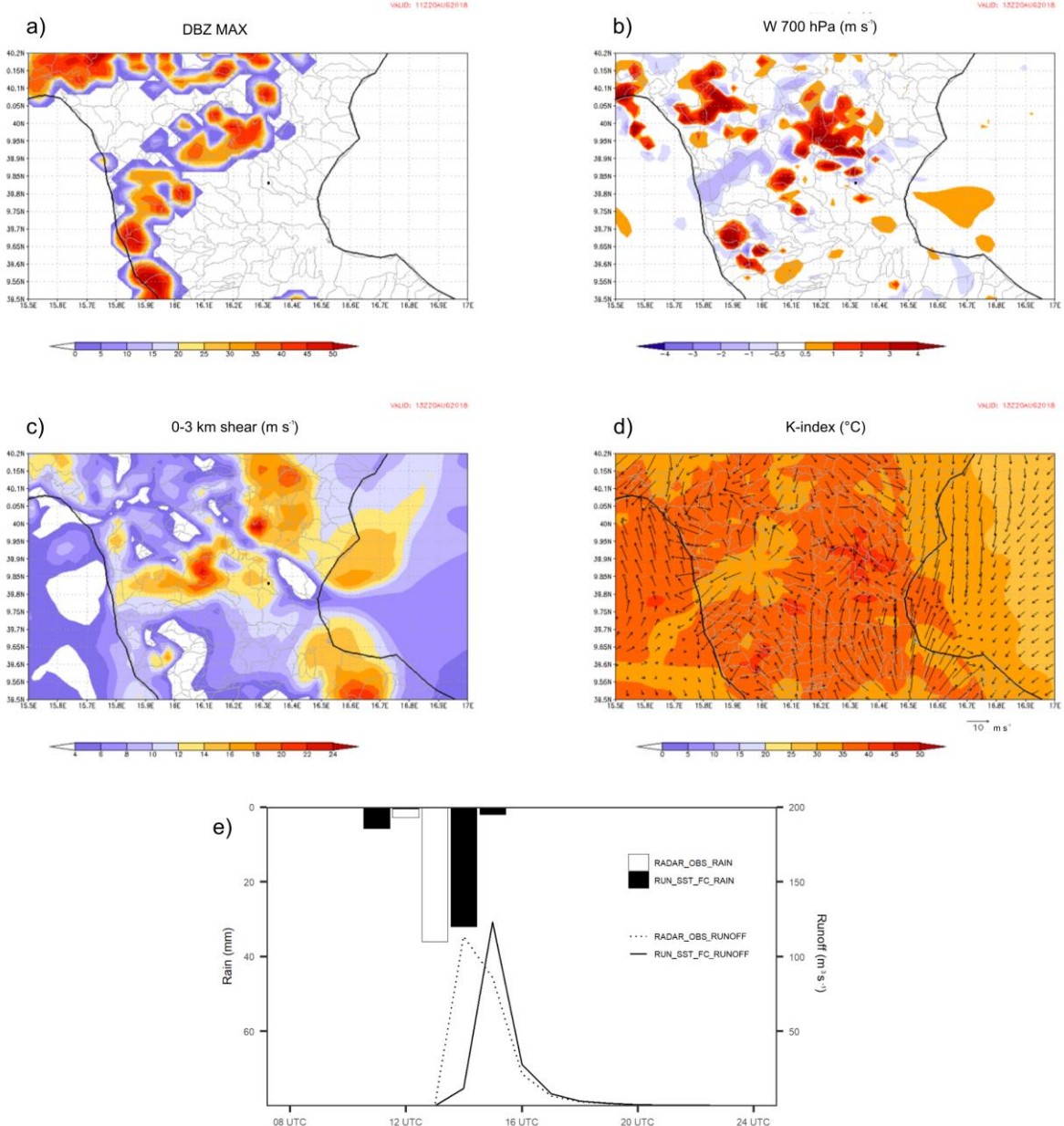

**Fig. 3: a) maximum reflectivity simulated at 11 UTC (dBZ); b) 700 hPa vertical wind speed simulated at 13 UTC (m s$^{-1}$); c) 0-3 km wind shear simulated at 13 UTC (m s$^{-1}$); d) k-index simulated at 13 UTC (°C); e) hourly averaged rainfall values over the catchment (mm) and the resulting hydrograph (m$^3$s$^{-1}$) achieved for radar-driven simulations and RUN_SST_FC. Results shown in Figs. from a) to d) are achieved with the run RUN_SST_FC.**