# Peer review of "This supplement is organized as follows: in the paragraph S1, Figs. S1 to S3 are related to the synoptic analysis. Then, in the paragraph S2 Table S1 shows the list of the preliminary tests carried out in order to achieve the first optimal configuration, while accompanying text, included Fig. S4, pr"

_Natural Hazards and Earth System Sciences, 2019_

## Referee Comment (RC1) · Anonymous Referee #1 · 5 Apr 2019

The manuscript analyzes an extreme rainfall-discharge event for a small ungauged mountainous catchment in southern Italy. The study describes in detail the synoptic preconditions that finally lead to the genesis of a strong convective precipitation event over orographically complex and steep terrain. Further, it is looked into how well such a local, short term event can be predicted by a dynamic hydrometeorological modeling system both for precipitation and discharge.

The manuscript is well structured, and gives a profound analysis of the available observations. The study fits in general to the objectives of the journal. The text is well written.

[Figure]

I have the following comments

Major

1) Please provide some more detail on the radar data that you use in the comparison

   1) What kind of processing is done to get precipitation from reflectivity and how well do the derived values fit to the existing observations at the nearby precipitation stations?

   2) For the discussion: might a combined product, merged from rain gauges and radar give a better base for analysis, is this planned to be included for further investigation?

2) You do an interesting model setup intercomparison of coupled vs. non-coupled and variant surface boundaries vs. invariant but what is lacking is a comparison of the resulting hydrograph (figure 3) with one that has been modeled using the radar derived precipitation as input. One could disaggregate the SRT field with the assumption that a large portion of total aggregated rainfall occurred within the hour before the flood event.

3) You speak of "reasonably well" for the model to simulate the radar observed precipitation field. To me it doesn't look so well, when looking to Fig. 2. With the fully-coupled SST configuration the amount of precipitation with respect to the surrounding rain gauges improves but the important peak (towards the east), as shown by the radar image on Fig. 2 is completely missed. In Figure 2 it is hard to locate the boundaries of the Raganello catchment, the black dot might be the outlet (but no description given in the figure caption), you should remove the administrative boarders and print the catchment boundary instead. It seems that WRF initiates convection in a different location thus leading to the mismatch in

positioning of the convective cell. Improved soil moisture initalization may help here. The simulation shows reasonable (but not satisfactory as stated in the abstract) results, but with a displaced / underestimated precipitation field and a decreased infiltration parameter.

4) In the following you reduce infiltration in the model (by setting REFKDT to 1.5) which leads to a higher discharge peak, but with the higher precipitation amount seen by the radar also the original value of REFKDT = 3 could have led to similar outcome. That corroborates the need for a radar-driven simulation. How does the bias look like for the 10-13 UTC basin aggregated sums precipitation sums? To analyze this, the few pixels in Raganello creek catchment can be manually digitized to get the necessary information.

5) This study does only an analysis of a single event. With the given configuration a lead time of 6 hours might have been achieved but this does not mean that the configuration would perform likewise for other events. To provide reliable forecasts, the system needs to be tested / set-up / evaluated for a broader range of events. This should also be honestly discussed on p7 around L10.

6) How do you plan to use radar information with 3d-Var? Do you plan to assimilate the reflectivity?

Minor

- Why it is gorge in the title and creek elsewhere?

- Figure 1c, highlight the catchment with different color or linetype than the administrative boundaries

- For comparison with other studies and model setups, I would like to see also information about catchment size and typical land-cover in the basin description; is the catchment intermittent or is there a more or less constant baseflow?

- Fig.3e, the unit for the precipitation amount should rather be mm instead of mm h-1 as it is a sum and not an intensity

- Fig.1 description L8, blank between dot and "The" missing and catchment or watershed may fit better to the extent of the Raganello Creek than basin (should be replaced throughout the manuscript)

P2L8 small-scale events instead of event

---

## Referee Comment (RC2) · Anonymous Referee #2 · 17 Apr 2019

The manuscript is well written and nicely shows the potential of WRF-Hydro for flash flood forecasting. Still, I have some comments which should be considered before publication in nhess.

(Major) comments

Page 4, line 11 to line 26: It seems that no spinup has been applied to generate the initial condition of land surface variables, including soil moisture. This may be problematic for a WRF-Hydro application, as a too wet soil condition in the forcing data may cause a discharge peak artifact at the beginning of the simulation. In the case of Senatore et al. (2015) a two-month spinup time was used. Please discuss this issue in

the revised version.

Page 5 line 23 to page 6 line 7: the comparison between the different model configuration results would be much more powerful in the framework of a model ensemble. The authors could generate a small ensemble based for example on randomly perturbed initial soil moisture condition, and assess how robust the differences between the model configurations are.

Minor comment

page 4, line 8: I suggest to replace "the diurnal cycle of Sea Surface Temperature" by "Sea Surface temperature dynamics", unless the author confirm that their input data provides subdaily variation of SST.

---

## Author Comment (AC1) · 14 May 2019

**We warmly thank the referees for their comments and their careful reading of our paper. Please find below our answers to all items raised (bold text).**

\* Major \*

1) Please provide some more detail on the radar data that you use in the comparison
a) What kind of processing is done to get precipitation from reflectivity and how well do the derived values fit to the existing observations at the nearby precipitation stations?

**Such as pointed out in the manuscript, we only had the opportunity to refer**

(thanks to the report published by the "Centro Funzionale Multirischi" of the Calabrian Regional Agency for the Protection of the Environment) to a (static) image of the SRT (surface rainfall total) product, processed by the Italian Civil Protection Department. Therefore, no detailed comparison with nearby rain gauges was possible. However, due to the fact that the monitoring network was not able to detect the event (most of the rain gauges recorded no rain), a possible evaluation would be substantially limited to the assessment of presence/absence of rain.

Radar observations are processed by the Italian Civil Protection Department according to nine steps detailed in Vulpiani et al. (2014), Petracca et al. (2018) and references therein. According to referee's comment, in the revised text we will explicitly (but briefly, according to the request of conciseness for Brief Communications) refer to that processing. Instead, a more detailed answer to the referee's comment is provided below.

Contamination by non-weather returns (clutter), partial beam blocking, attenuation at increasing distance, vertical variability of precipitation are the main sources of error for the radar observations. The cited references adopt a specific approach for radar data quality estimation to compensate such sources of error.

The algorithm mainly consists in retrieving the mean Vertical Profile of Reflectivity (VPR) and computing the Surface Rainfall Intensity (SRI) maps (both VPR and SRI are corrected through data quality control). Then, the Surface Rainfall Total (SRT, aka accumulated rainfall) is achieved by integrating SRI maps.

SRI is calculated applying a Z–R (reflectivity–rainfall intensity) relationship; the default assumption is that proposed by Marshall and Palmer (1948): $Z=200*R^{1.6}$.

In specific cases (orographically complex areas) the authors found that the use of radar reflectivity for estimating precipitation is frequently subject to underes-

timation, mainly due to orographic features. In order to reduce this effect, an alternative method is to derivate the SRI from the VMI (Vertical Maximum Intensity), ground-projected by means of the retrieved VPR (Rinollo et al., 2013).

In the cited references, performance analysis procedures were carried out, using the rain gauges as benchmark; from the comparisons, rainfall estimation accuracy resulted relatively high.

b) For the discussion: might a combined product, merged from rain gauges and radar give a better base for analysis, is this planned to be included for further investigation?

In our work we already considered both the rain gauges and a radar product (please refer to P2L17 of the Manuscript). Specifically, we used the SRT image to firstly discriminate the configurations that better simulate the precipitation patterns (P3L19) and the rain gauges measurements to perform a basic quantitative statistical analysis aimed at choosing the best configuration (P5L30). We will strive to better highlight this concept in the revised manuscript, envisaging the possible use also of combined products. However, in situations like the case study here analysed, the high localization of the event (most of the rain gauges recorded no rain) makes more difficult to successfully apply a merging procedure, even with fully available radar data.

2) You do an interesting model setup intercomparison of coupled vs. non-coupled and variant surface boundaries vs. invariant but what is lacking is a comparison of the resulting hydrograph (figure 3) with one that has been modeled using the radar derived precipitation as input. One could disaggregate the SRT field with the assumption that a large portion of total aggregated rainfall occurred within the hour before the flood event.

We thank the referee for appreciating the model setup intercomparison and for the comment. We have followed his/her suggestion, even though acknowledging the related levels of uncertainty, and added a new resulting hydrograph in a new figure (we will replace the Fig. 3e in the revised paper), which is shown

**in our response to next point 4. The radar-derived precipitation has been transformed in WRF-Hydro input assigning to each pixel of the radar image a rainfall value equal to the central value of the corresponding colour class, then pixels have been aggregated from radar (1 km) to WRF (2 km) resolution by calculating the average. Temporal disaggregation has been made in proportion to the WRF output.**

3) You speak of "reasonably well" for the model to simulate the radar observed precipitation field. To me it doesn't look so well, when looking to Fig. 2. With the fully-coupled SST configuration the amount of precipitation with respect to the surrounding rain gauges improves but the important peak (towards the east), as shown by the radar image on Fig. 2 is completely missed.

**The main reason for the non-perfect agreement between the 10-13 UTC SRT image on Fig. 2a (which, it's worth it to recall, is the only available for comparison) and model results (Figs. 2b-e) is related to the acknowledged small delay in the rainfall simulation (P5L11-L16). In the manuscript, we first refer to the "reasonably well" simulated "C" shape of the rain pattern (P5L11), then we speak about the system ability to simulate "reasonably well" the rain field (P7L28), which we believe is an acceptable statement if the abovementioned delay is taken into account. Unfortunately, the main precipitation peak is not correctly located in all simulations (from 15 to 17 km with respect to radar estimates, as we acknowledged in the text).**

**In the revised text we will strive to make this shortcoming more clear.**

In Figure 2 it is hard to locate the boundaries of the Raganello catchment, the black dot might be the outlet (but no description given in the figure caption), you should remove the administrative boarders and print the catchment boundary instead.

**We thank the referee for this suggestion; we will modify both Figure 2 and the caption accordingly.**

It seems that WRF initiates convection in a different location thus leading to the mismatch in positioning of the convective cell. Improved soil moisture initialization may help here.

**Such as pointed out in the manuscript (P8L11), future work will be performed in order to evaluate the possible role of soil moisture initialization. Also other causes can concur to the mismatch (e.g., too much smoothed elevation and slope of the mountain chain; effects of elevation representation will be also explored). However, since the topic of soil moisture initialization is addressed by both referees (please refer also to our response to referee 2 first comment), we were motivated to perform a preliminary test running a simulation with ERA5 reanalysis boundary conditions for the two weeks preceding the event; the soil moisture fields achieved were then used as initial conditions for our simulation. Preliminary results shown in Fig. 1 in this reply (please find it at the end of this document) seem to suggest that improved soil moisture initialization alone is not able to improve convective cell positioning. The issue of improved soil moisture initialization will be discussed with more details in the revised manuscript.**

The simulation shows reasonable (but not satisfactory as stated in the abstract) results, but with a displaced / underestimated precipitation field and a decreased infiltration parameter.

**We agree with referee's comment about displacement/underestimation errors and will modify both the Abstract and the Results section accordingly.**

**Concerning the decreased infiltration parameter, such as we try to explain in the next point 4, it is not a matter of underestimated precipitation fields, rather it is related to differences in the calls to the NOAH vertical infiltration scheme between the one-way and fully-coupled versions of WRF-Hydro.**

4) In the following you reduce infiltration in the model (by setting REFKDT to 1.5) which leads to a higher discharge peak, but with the higher precipitation amount seen by the

radar also the original value of REFKDT = 3 could have led to similar outcome. That corroborates the need for a radar-driven simulation. How does the bias look like for the 10-13 UTC basin aggregated precipitation sums? To analyze this, the few pixels in Raganello creek catchment can be manually digitized to get the necessary information.

**The infiltration parameter REFKDT was reduced not with the aim of achieving a higher discharge peak (after all, this attempt would make no sense, because there is no observed peak for comparison), but because we took into account some intrinsic differences between one-way and fully-coupled versions of WRF-Hydro.**

**When WRF-Hydro is run in offline mode, the land model is called once per hour. In the fully coupled run, instead, the land model (and hence all the WRF-Hydro routines) is called on the WRF model physics time step, which is in the order of seconds. Such as Senatore et al. (2015) highlighted,** *"the difference in land model execution frequency is important because it impacts how frequently infiltration and other fluxes are calculated. If the land model is called infrequently, then routed waters can travel farther downslope or into a channel before infiltration happens again. When the land model is called frequently, infiltration is calculated more frequently and thus more water infiltrates rather than making it all the way into a channel. Hence, more frequent land model calls usually result in more infiltration and less runoff, meaning also lower peak flows"*.

**Therefore, REFKDT was reduced in order to compensate different intervals used for temporal integration of the Noah-LSM between the two versions of the model. This approach is supported also by the results achieved with the radar-driven simulation. Such as shown in Fig. 2 in this reply (which will be Fig. 3e in the revised paper), the one-way radar driven and the fully-coupled simulations have similar rainfall amounts in input. In order to get, such as expected, also similar discharge peaks, REFKDT is set equal to 3.0 (default value) in the one-way simulation and to 1.5 in the fully-coupled simulation.**

**We will strive to be more clear about this point in the revised manuscript.**

5) This study does only an analysis of a single event. With the given configuration a lead time of 6 hours might have been achieved but this does not mean that the configuration would perform likewise for other events. To provide reliable forecasts, the system needs to be tested / set-up / evaluated for a broader range of events in the region. This should also be honestly discussed on p7 around L10.

**We agree with referee's comment. To provide general indications about the lead-time, the forecasting system should be tested for a broader range of events. We will emphasize in the revised text that this evaluation is based on a single-case test.**

6) How do you plan to use radar information with 3d-Var? Do you plan to assimilate the reflectivity?

**Yes, we do. WRF allows to assimilate both radial velocity and reflectivity (direct and indirect assimilation). Specifically, our plan is to evaluate indirect assimilation of reflectivity, diagnosing microphysics and humidity parameters (from reflectivity), and assimilating these diagnosed quantities in the model. This approach permits to consider cloud and vertical velocity control variables.**

* Minor *

Why it is gorge in the title and creek elsewhere?

**Our intention was to identify, in the title, the area where the flood caused its maximum impact (i.e., the flash flood affected the downstream outlet of the gorge of the Raganello Creek Catchment), in order to refer to the event more clearly and directly. In the main text, except once in the summary, we rather refer to the Raganello Creek Basin (the word Basin will be replaced with Catchment, according to a comment following). We would prefer to maintain this terminology.**

Figure 1c, highlight the catchment with different color or linetype than the administrative

boundaries.

**Thanks for this suggestion, we will change the figure accordingly (thicker catchment contours and light grey administrative borders).**

For comparison with other studies and model setups, I would like to see also information about catchment size and typical land-cover in the basin description; is the catchment intermittent or is there a more or less constant baseflow?

**Thanks for this suggestion, which reminds us that fundamental information is missing. We will add it in the revised version of the manuscript. Catchment extent is about 100 km2. According to Corine Land Cover 2018 inventory, almost half of the land is covered by forest (44%, almost all broad-leaved forest), 22.9% by shrubs, 21.8% by agricultural areas (13.8% heterogeneous agricultural areas, 7.7% non-irrigated arable land), 11% by open spaces with little or no vegetation. Artificial surfaces are only 0.3%. Even though nor discharge neither water level monitoring stations are available at the analysed catchment outlet, it can be reasonably stated that the creek in that section is perennial (during the whole dry summer season it is a destination for tourists who practice canyoning).**

Fig.3e, the unit for the precipitation amount should rather be mm instead of mm h$^{-1}$ as it is a sum and not an intensity

**Thanks for this suggestion. We will change the figure accordingly.**

Fig.1 description L8, blank between dot and "The" missing and catchment or watershed may fit better to the extent of the Raganello Creek than basin (should be replaced throughout the manuscript)

**Thanks for these suggestions. We will add blank space. The word "basin" will be replaced throughout the manuscript with the word "catchment".**

P2L8 small-scale events instead of event

**Thanks for this suggestion. We will modify the text accordingly.**

**References**

**- Marshall, J. S. and Palmer, W. M.: The distribution of raindrops with size, J. Meteorol., 5, 165–166, 1948.**

**- Petracca, M., L. P. D'Adderio, F. Porcù, G. Vulpiani, S. Sebastianelli, and S. Puca, 2018: Validation of GPM Dual-Frequency Precipitation Radar (DPR) rainfall products over Italy. J. Hydrometeor., 19, 907–925, https://doi.org/10.1175/JHM-D-17-0144.1**

**- Rinollo, A., Vulpiani, G., Puca, S., Pagliara, P., Kaňák, J., Lábó, E., Okon, Ľ., Roulin, E., Baguis, P., Cattani, E., Laviola, S., and Levizzani, V.: Definition and impact of a quality index for radar-based reference measurements in the H-SAF precipitation product validation, Nat. Hazards Earth Syst. Sci., 13, 2695-2705, https://doi.org/10.5194/nhess-13-2695-2013, 2013.**

**- Vulpiani, G., A. Rinollo, S. Puca, and M. Montopoli, 2014: A quality-based approach for radar rain field reconstruction and the H-SAF precipitation products validation. Proc. Eighth European Radar Conf., Garmish-Partenkirchen, Germany, ERAD, Abstract 220, 6 pp., http://www.pa.op.dlr.de/erad2014/programme/ExtendedAbstracts/220_Vulpiani.pdf (last access January 2019).**
* * *
[Figure]

**Fig. 1.** On the left, Fig. 1c of the manuscript (24h accumulated precipitation according to the run RUN_SST_FC); on the right, the same but with improved soil moisture initialization.

**Fig. 2.** Hourly averaged rainfall values over the catchment and resulting hydrographs for radar-driven simulations and RUN_SST_FC.

---

## Author Comment (AC2) · 14 May 2019

**We warmly thank the referees for their comments and their careful reading of our paper. Please find below our answers to all items raised (bold text).**

(Major) comments Page 4, line 11 to line 26: It seems that no spinup has been applied to generate the initial condition of land surface variables, including soil moisture. This may be problematic for a WRF-Hydro application, as a too wet soil condition in the forcing data may cause a discharge peak artifact at the beginning of the simulation. In the case of Senatore et al. (2015) a two-month spinup time was used. Please discuss this issue in the revised version.

[Figure]

The referee is right. In our study, initial and boundary conditions were provided directly by ECMWF-IFS in its deterministic forecast version at 9 km resolution. We acknowledge the importance of proper spin-up and we plan to study this issue with more detail (P8L11 in the manuscript: "future work will be devoted to the improvement of soil moisture initial conditions, through a dedicated hydrological analysis assuring a proper spin up time"). According to referee's suggestion, we will discuss with more detail (even though briefly, in compliance to the request of conciseness for Brief Communications) the issue in the revised version of the manuscript, summarizing the discussion provided in the following.

We believe that, in the framework of a Brief Communication, the choice made about initial conditions is consistent with the objective of providing preliminary and early indications for operational hydro-meteorological forecasting purposes in the study area. Proper spin-up for soil moisture can be done operationally (actually, it is currently done in some systems) through off-line seamless run of the hydrological model fed by observations provided in real time, but this procedure is feasible only for the innermost domain and requires that the forecasting system is expressly set-up with several non-trivial procedures (e.g., real-time seamless data transmission and validation from the monitoring network, real-time spatial interpolation of punctual observations and production of input fields, possible data assimilation procedures, replacement of the land surface initial conditions in the model, etc.). Senatore et al. (2015) used a two-month spin-up strategy, for a 3-year simulation period. In that case, no operational (short-term) forecast purposes were pursued and ERA-Interim reanalysis was used.

Since the topic of soil moisture initialization is addressed by both referees (please refer also to our response to referee 1 third major comment), we were motivated to perform a preliminary test running a simulation with ERA5 reanalysis boundary conditions (of course, this product is not usable operationally) for

**the two weeks preceding the event. The soil moisture fields achieved were then used as initial conditions for our simulation. Preliminary results shown in Fig. 1 in this reply (please find it at the end of this document) seem to suggest that improved soil moisture initialization, alone, is not able to improve convective cell positioning. The accumulated rainfall fields over the catchment are quite similar, such as the resulting hydrographs. Anyway, it should be highlighted that the slight difference between discharge forecasts should be attributed to both the differences in the rainfall fields and in initial soil moisture conditions, therefore the effect of the latter cannot be isolated. Furthermore, even though isolated, their added value over the hydrological simulation could not be appreciated in this specific case, due to the lack of discharge observations.**

Page 5 line 23 to page 6 line 7: the comparison between the different model configuration results would be much more powerful in the framework of a model ensemble. The authors could generate a small ensemble based for example on randomly perturbed initial soil moisture condition, and assess how robust the differences between the model configurations are.

**The issue of ensemble/probabilistic forecasting is certainly of paramount importance. We thank the referee for highlighting this point, that will be introduced and discussed in the revised version of the manuscript. However, it is worth it to recall that many ensemble approaches can be adopted, concerning both the atmospheric and hydrological modelling compartments. In this work, our main goal is to evaluate the modelling forecasting skills for early warning purposes, trying to minimize the lead-time with which forecasts are provided (this issue is particularly important for small, highly responsive catchments), in order to set-up a possible operational forecasting system devoted to hydrological risk assessment. Clearly, an operational procedure including a number of ensemble simulations would significantly penalize the calculation/processing speed of the forecast results, with consequent increase of lead-time, if not adequately sup-**

[Figure]

**ported by heavy computational resources.**

**Concerning the ensemble experiment proposed by the referee in order to "assess how robust the differences between the model configurations are", our opinion is that it, though interesting, would provide only a partial picture of the probability forecasting, especially in the framework of a Brief Communication. Therefore, we would prefer to avoid treating this topic in the manuscript. Nevertheless, intrigued by the suggestion of the referee, we have already performed some preliminary tests. Specifically, we have perturbed initial soil moisture conditions of simulations RUN_SA (the first identified optimal configuration) and RUN_SST_FC (the best performing configuration), considering in both cases uniform changes of $\pm$5%.**

**Preliminary results show interesting points. E.g., for RUN_SA, 5% increased initial soil moisture provides enough water to the system, so that the rainfall peak value increases up to a value comparable to RUN_SST_FC. On the other hand, with RUN_SST_FC rainfall increase/decrease is not so straightforwardly correlated to soil moisture increase/decrease. These results, within the framework of a comprehensive ensemble forecast, will be considered for further improvements of the forecasting system.**

Minor comment

page 4, line 8: I suggest to replace "the diurnal cycle of Sea Surface Temperature" by "Sea Surface temperature dynamics", unless the author confirm that their input data provides subdaily variation of SST.

**Thanks for this suggestion. We will change the text accordingly in the revised manuscript.**

[Figure]

**Fig. 1.** On the left, Fig. 1c of the manuscript (24h accumulated precipitation according to the run RUN_SST_FC); on the right, the same but with improved soil moisture initialization.

---

## Author Response (AR2)

We would like to thank once more the editor and the referee for their careful reading of the manuscript and their positive comments. All the final suggestions have been accepted (please find below the manuscript with tracked changes). Furthermore, interacting with our colleague Linus Magnusson (ECMWF), we decided to slightly modify his personal communication, from "*
[revised manuscript text omitted]